# Compass Cue Integration and Its Relation to the Visual Ecology of Three Tribes of Ball-Rolling Dung Beetles

**DOI:** 10.3390/insects12060526

**Published:** 2021-06-06

**Authors:** Lana Khaldy, Claudia Tocco, Marcus Byrne, Marie Dacke

**Affiliations:** 1Lund Vision Group, Department of Biology, Lund University, Sölvegatan 35, 223 62 Lund, Sweden; claudia.tocco@biol.lu.se (C.T.); marie.dacke@biol.lu.se (M.D.); 2School of Animal, Plant and Environmental Sciences, University of the Witswatersrand, 1 Jan Smuts Avenue, Braamfontein, Johannesburg 2000, South Africa; marcus.byrne@wits.ac.za

**Keywords:** orientation, orientation strategy, visual ecology, dung beetle, compass cues

## Abstract

**Simple Summary:**

To escape competition for food at the dung pat, ball-rolling dung beetles shape a piece of dung into a sphere and roll it away. To maintain their bearing, these beetles integrate directional information from a range of celestial cues. For the majority of diurnal dung beetles living in open habitats, the most dominant of these cues is the sun. It has recently been demonstrated that beetles living in closed habitats, with closely spaced trees and tall grass, rely predominantly on directional information provided by polarised skylight rather than the sun. Taken together, these findings suggests that the orientation strategy of the beetle is influenced by the animal’s visual ecology. To further investigate the relative weighting of cues in the orientation system of beetles, and its relation to their visual ecology, we investigated the orientation strategy of ball-rollers from three different dung beetle tribes, all present within the same savanna biome. We find that species within a tribe share the same orientation strategy, but that this strategy differs across tribes. We conclude that, despite dramatic intertribal differences in body size and external eye design, the dynamic heading direction network of the South African ball-rolling dung beetles is well adapted to guide the foraging insect in the habitat that it normally traverses.

**Abstract:**

To guide their characteristic straight-line orientation away from the dung pile, ball-rolling dung beetles steer according to directional information provided by celestial cues, which, among the most relevant are the sun and polarised skylight. Most studies regarding the use of celestial cues and their influence on the orientation system of the diurnal ball-rolling beetle have been performed on beetles of the tribe Scarabaeini living in open habitats. These beetles steer primarily according to the directional information provided by the sun. In contrast, *Sisyphus fasciculatus*, a species from a different dung-beetle tribe (the Sisyphini) that lives in habitats with closely spaced trees and tall grass, relies predominantly on directional information from the celestial pattern of polarised light. To investigate the influence of visual ecology on the relative weight of these cues, we studied the orientation strategy of three different tribes of dung beetles (Scarabaeini, Sisyphini and Gymnopleurini) living within the same biome, but in different habitat types. We found that species within a tribe share the same orientation strategy, but that this strategy differs across the tribes; Scarabaeini, living in open habitats, attribute the greatest relative weight to the directional information from the sun; Sisyphini, living in closed habitats, mainly relies on directional information from polarised skylight; and Gymnopleurini, also living in open habitats, appear to weight both cues equally. We conclude that, despite exhibiting different body size, eye size and morphology, dung beetles nevertheless manage to solve the challenge of straight-line orientation by weighting visual cues that are particular to the habitat in which they are found. This system is however dynamic, allowing them to operate equally well even in the absence of the cue given the greatest relative weight by the particular species.

## 1. Introduction

To successfully navigate the world, animals often rely on directional information from more than one sensory channel [1,2,3]; fruit flies require visual feedback to localize an odour source [4], bees are better at pin-pointing their nest entrance with the addition of olfactory cues [5] and Bogong moths use the Earth’s magnetic field in combination with landmarks to localize Alpine caves [6]. In addition, orienting insects seem to afford the greatest weight to the directional information that conveys the highest certainty at a given moment [7]. Monarch butterflies primarily rely on the sun to find their route across the North American continent [8] but will refer to polarised skylight for directional guidance as soon as this bright solar cue is obstructed [9,10,11], and as the sun climbs high in the sky, becoming less reliable for directional input, dung beetles and ants rely more heavily on directional information provided by wind [1,12]. Homing ants, which find their way back to their nest by path integration and landmarks, also employ a dynamic strategy for reliable navigation; as these foragers are displaced further and further from their nest and the visual scenery around them becomes increasingly unfamiliar, they shift the relative directional weight of their path integrator and landmark guidance in favour of the former to lead them back home [13].

Not surprisingly, the visual ecology of an animal influences what directional cues to follow where and when. Rodent and fish species living in spatially complex environments will rely more on egocentric cues to find their way compared to species inhabiting more open habitats [14,15,16]. Another example can be found among ants, where species inhabiting cluttered, landmark-rich spaces, rely more on landmark guidance compared to desert ants that forage in open, featureless habitats [13,17,18,19,20]. While these differences in directional guidance appear to be species specific, and strictly tuned to the visual environment in which the animal lives, a dynamic influence of the visual ecology of the navigator can be observed in the strictly nocturnal, savanna-living dung beetle, *Scarabaeus satyrus* (Fabricius). During a moon-lit night, this beetle orients using polarised lunar skylight in preference to the moon, but if coerced to roll during the day, the heading direction network of the beetle shifts the relative weight of these two types of celestial cues in favour of directional information provided by the sun [21].

Diurnal ball-rolling dung beetles steer their characteristic straight-line escapes from a dung pile [22,23,24,25,26,27] by directional information provided by the sun [21,23,24,27], the polarised skylight [26,28], the gradients of intensity [28] and colour that form across the daytime sky [29], as well as the prevailing winds [1]. Our understanding of how these insects roll straight over the sun-lit savanna is largely based on behavioural, morphological and neurobiological studies of beetles from the tribe Scarabaeini [1,21,23,27,28,30] (but see [24,26,31] and below). In these studies, we repeatedly found that if the directional information from the sun is set in conflict with other celestial cues, the beetles change their bearings according to the position of the sun [23,30]. These experiments clearly demonstrate that directional information from the sun is given the greatest relative weight during straight-line orientation in these large and iconic dung beetle species. In addition, it seems that, contrary to homing ants, which choose an intermediate route when directional cues are set in conflict [12,32,33,34,35,36], dung beetles do not average the dictates of the directional sources, but instead predominantly rely on the directional information given the greatest weight at that time. Consequently, only when the sun is hidden from view will the Scarabaeini beetles turn in response to the rotation of an overhead pattern of polarised light [28].

It was recently shown that a savanna woodland-living species of the tribe Sisyphini, *Sisyphus fasciculatus*, displays a different behaviour; when rolling under a polarising filter under a sun-lit sky outdoors, this beetle changes its bearing in accordance with the turn of the polariser [26]. This suggests that these small beetles, which traverse litter strewn terrain under closely spaced trees or through tall grass, predominantly rely on directional information from the celestial pattern of polarised light [26]. Due to this contrasting behaviour, Khaldy et al. [26] suggested that the visual ecology of the orientation system of the different species of dung beetle, just as in ants, is influenced by their distinct habitat associations. Here, we continue to explore the relative weight of directional information in the heading direction network of three species of dung beetles from three different tribes, foraging in the closed or open habitat of the same savanna biome.

## 2. Materials and Methods

### 2.1. Selection of Dung Beetle Species

The three species of ball-rolling dung beetles initially included in this study all occur in the savanna biome [37]: *Kheper nigroaeneus* [tribe Scarabaeini], *Garreta unicolor* (tribe Gymnopleurini) and *Sisyphus fasciculatus* (tribe Sisyphini). The addition of *G. nitens* (tribe Gymnopleurini) as a fourth test species from the same biome was inspired by the unexpected finding that the heading direction network of *G. unicolor* did not attribute the greater directional weight to either the sun or the polarised light pattern (see Section 3.6). The experiments performed with this species is thus limited to defining the relative weight of directional cues in its orientation system.

### 2.2. Collection and Maintenance of Animals

Beetles were collected using dung-baited pit-fall traps in the Wits University, Pullen nature reserve (closed and open habitat) (31.10° E, 25.34° S) (*Kheper nigroaeneus*, *Garreta unicolor*, *Sisyphus fasciculatus*) and Bersig Eco Estate (open habitat) (27.95° E, 24.78° S) (*Garreta nitens*), South Africa. For illustrative purposes, the sampled habitats were photographed from the air (DJI Mavic 2) and from the ground (Nikon D810 fitted with an 8 mm fisheye lens) (see Figure 1). Once collected, beetles were maintained outside, in soil-filled, transparent plastic bins, and fed with fresh cow dung every second day. Beetles taken to the Department of Biology, Lund University, Sweden, were housed in large plastic bins (50 × 36 × 27 cm) in a light- and temperature-controlled room, under a 12 h light/dark cycle at a room temperature of 26 °C.

### 2.3. Determining Habitat Preference and Eye Size of the Dung Beetles

#### 2.3.1. Habitat Preference

To determine the habitat preferences of *Kheper nigroaeneus*, *Garreta unicolor* and *Sisyphus fasciculatus*, pitfall-traps were placed in the open habitat (dominant grass species: *Heteropogon contortus*, *Sporobolus pyramidalis* and *Chloris pycnothrix*) (Figure 1a) and closed habitat (dominant tree species: *Sclerocarya birrea, Searsia pentheri* and *Erythrina lystemon*) (Figure 1b) for three non-consecutive sampling sessions during March 2019. Traps were emptied and re-baited with fresh dung every 3 h during daylight hours. For more details regarding the trapping method, see Khaldy et al., 2020 [26]. The habitat preference for *G. nitens*, that shares the same savanna biome, was not defined.

#### 2.3.2. Statistical Analysis of Habitat Preference

To test for differences in species abundance between habitat types, generalized linear mixed models (GLMMs) [38] in R (R Core Team 2020, Vienna, Austria, https://www.R-project.org/, accessed on 31 May 2021), used with *lme4* [39], were fitted. Each trap of each sampling event was used as a sampling unit, with a total of 155 sampling units. The Shapiro–Wilk test was used to test for normality in the residual distribution of the species abundance. The abundance of each species was non-normal count data and Poisson error distribution was specified in each model [39]. In all GLMMs, habitat type was treated as a fixed factor and sampling day as a random factor to block the layout of the sampling design.

#### 2.3.3. Eye Size

To measure the eye surface area, the right eye of ten individuals of each species was covered with a thin layer of transparent nail polish. Once dried, the coat of nail polish was peeled off from the eye, cut and mounted flat on a microscope slide. The images of the flattened impression of the eyes were taken with a stereo microscope (Zeiss Stereo Discovery V12) and the absolute area was measured using ImageJ (Rasband, W.S., ImageJ, U. S. National Institutes of Health, MD, USA, https://imagej.nih.gov/ij/, 1997–2018, accessed on 20 May 2021). As *K. nigroaeneus* possesses a complete canthus, the absolute eye area for this species was calculated as the sum of the dorsal and ventral eye area.

### 2.4. Behavioural Experiments

Outdoors, experiments were performed under clear skies, at solar elevations between 45° and 60°, at Bersig Eco Estate and Pullen nature reserve between March 2018 and November 2019. In Lund University, Sweden, the beetles were presented with a green unpolarised light spot (Adafruit DotStar Digital LED Strip; emission peak 530 nm, Adafruit Industries, New York, NY, USA), a previously documented replacement for the sun in the heading direction network of the beetle [21], at an elevation of 45°, in an otherwise completely darkened indoor room. An overhead Sony Handycam HDR-CX730E (fitted with a 0.42× wide angle lens), mounted from above with the lens facing downwards, was used to record exit bearings.

#### 2.4.1. Orientation Performance of Dung Beetles

To determine the beetle’s orientation performance under an open sky, each individual was repeatedly placed beside its ball in the centre of a circular, flat, sand-coated arena, where the effective radius was set to a distance equivalent to the length of 20 steps for the species tested (*K. nigroaeneus*; 59 cm, *G. unicolor*; 32 cm, *S. fasciculatus*; 32 cm) (for detailed data see Appendix A). Each beetle was allowed to roll its ball to the arena perimeter ten times. Ten individuals per species were tested.

#### 2.4.2. Relative Weighting of Directional Cues in the Orientation System of Dung Beetles

For each experimental treatment, the beetle was placed alongside its dung ball, in the centre of a 50 cm radius circular arena and allowed to roll its ball to the perimeter where its exit bearing was noted. For conditions requiring a polarising filter, a circular 30 cm radius, UV/Visible light-transparent polarisation filter (BVO UV Polarizer, Bolder Vision Optik©, Boulder, CO, USA) was positioned over the centre of the arena. The filter was mounted on four legs (10 cm in height) and fitted with a black cloth curtain around its perimeter to prevent the entry of light from outside the filter. The exit bearing was recorded when the beetle reached the filter perimeter. Upon completion of the beetle’s first roll, the position of the test cue(s) was rotated by either 90° or 180° (see Section 2.4.3. below). The beetle was allowed to exit the arena and its second exit bearing was noted. A third exit, presenting the same visual parameters as in the first trial, was performed as a control to test whether the beetle could follow approximately the same bearing throughout the experiment. Angular change was calculated as the difference in bearing between the first and second exit (*test*), or first and third exit (*control*). In total, each individual rolled from the centre to the edge of the arena (or filter perimeter) three times. In all outdoor experiments, 20 individuals per species were tested. For the indoor experiments, 10 individuals were tested for each species.

#### 2.4.3. Manipulation of Directional Input

*Sun (ersatz or real):* In the field, the sun’s apparent position was changed by 180° using a mirror (30 × 30 cm), while simultaneously concealing the real sun from the beetle’s field of view using a wooden shade board (100 × 75 cm). Indoors, the azimuth of the ersatz sun was changed by 180° between trials by switching off and on the green light spot at different relative positions.

*Polarised light:* In the field, the UV/Visible light-transparent polarisation filter was turned by 90°, between consecutive rolls, either under a full view of the sun or with the sun shielded from the beetle’s field of view by the shade board. The initial orientation of the filter was alternated for each beetle, with every second beetle starting with the polarisation filter aligned to the natural polarisation band of the sky, and every other beetle with the filter aligned perpendicular to the natural polarisation band of the sky.

*Sun and polarised light:* In these experiments, the polarising filter was turned by 90° in combination with a 180° change in the solar position, as described above.

#### 2.4.4. Circular Statistics

Circular statistics on measured data was performed using Oriana 4.0 (Kovach Computing Services, Anglesey, UK). All circular data are reported as mean ± one circular standard deviation. Distributions of exit angles were analysed using Rayleigh’s uniformity test for circular data [40]. Changes in direction between treatments were calculated by measuring the angular difference in exit bearing between two exits from the arena and analysed using a v-test with an expected mean of 0° for the control experiments and 180° for the mirrored sun/ersatz sun experiments. To test for homogeneity of two or more samples, a Mardia–Watson–Wheeler test was used.

## 3. Results

### 3.1. Habitat Preference

*Kheper nigroaeneus* and *G. unicolor* were primarily found actively foraging within the open habitat (open vs. closed habitat: *K. nigroaeneus*; *p* < 0.001, z-value = 8.60, estimate = 2.09, N = 165; *G. unicolor*; *p* < 0.001, z-value = 18.41, estimate = 1.55, N = 971, GLMM test) (Figure 1a, histogram), while *S. fasciculatus* was mainly found in the closed habitat (closed vs. open habitat: *S. fasciculatus*; *p* < 0.001, z-value = −19.39, estimate = −2.87, N = 939, GLMM test) of the same bioregion (Figure 1b, histogram). These findings strongly suggest that *K. nigroaeneus* and *G. unicolor* preferentially forage for dung in the open habitat, while *S. fasciculatus* forages for dung in the closed habitat of the same bioregion. 

### 3.2. Differences in Eye Size and Shape

The relatively big eye (1.60 ± 0.57 mm^2^) of *K. nigroaeneus* is completely divided into a dorsal (0.59 ± 0.22 mm^2^) and a ventral part (Figure 2a), while the smaller eyes of *G. unicolor* (0.21 ± 0.03 mm^2^) and *S. fasciculatus* (0.15 ± 0.03 mm^2^) rather have a more oval-shaped dorsal eye (0.05 ± 0.01 and 0.02 ± 0.01 mm^2^, respectively) which connects to the ventral part of the eye (N = 10) (Figure 2b,c) (for detailed data see Appendix A).

### 3.3. Orientation Performance under the Natural Sky Is Equal for All Species

The outdoor orientation performance of the three species, as determined from the mean resultant vector length (R) of 10 exit bearings per beetle from the centre of the circular arena (the closer to 1, the better oriented the beetle) did not differ between the species (*K. nigroaeneus*: R = 0.93 ± 0.1; *G. unicolor*: R = 0.93 ± 0.1; *S. fasciculatus*: R = 0.88 ± 0.1, *p* = 0.15, Kruskal–Wallis test, N = 10) (for detailed data see Appendix A). We also found that within a species, the first bearing chosen by each individual was not biased towards any particular heading (*K. nigroaeneus*: *p* = 0.06, Z = 2.8; *G. unicolor*: *p* = 0.54, Z = 0.64; *S. fasciculatus*: *p* = 0.22, Z = 1.53, Rayleigh uniformity test, N = 10). Although the evidence for this was weaker in *K. nigroaeneus* (*p* = 0.06), previous work on closely related Scarabaeini species [22,30] suggests that this is most likely an effect of the small sample size. Taken together, this indicates that, under an open sky, our test species, from three different tribes, are able to travel along any given bearing with the same angular precision.

### 3.4. Ball-Rolling Dung Beetles Can Orient to a Single Green Light Spot 

Beetles presented with a green light spot (indoors) as an ersatz sun in the same azimuthal position between two consecutive exits from the centre of the arena (*control*), and showed no significant change in direction in any of the three species, (*K. nigroaeneus*: μ = 28.37° ± 49.27°, *p* < 0.01, V = 2.72; *G. unicolor*: μ = 333.7° ± 54.70°, *p* < 0.01, V = 2.54; *S. fasciculatus*: μ = 5.13° ± 39.89°, *p* < 0.001, V = 3.50, mean ± circular s.d., v-test (with the expected mean of 0°), N = 10) (Figure 3a, grey dotted vector).

When the position of the ersatz sun was changed by 180° between two exits from the centre of the arena (*test*), all species changed their headings accordingly (*K. nigroaeneus*: μ = 219.59° ± 37.73°, *p* < 0.01, V = 2.77; *G. unicolor*: μ = 186.27° ± 49.40°, *p* < 0.001, V = 3.47; *S. fasciculatus*: μ = 185.9° ± 41.67°, *p* < 0.001, V = 3.41, v-test (with the expected mean of 180°), N = 10) (Figure 3a). These changes in headings showed that the species tested can steer with reference to a single point-light source and with no significant difference in performance between species (*p* = 0.139, W = 6.95, Mardia–Watson–Wheeler test, N = 10).

### 3.5. The Role of the Sun in the Orientation System of Ball-Rolling Dung Beetles

When allowed to exit the arena under the open sky, followed by an exit where the apparent solar position was mirrored by 180° (*test*), *K. nigroaeneus* still showed a marked change in heading (μ = 201.05° ± 69.46°, *p* < 0.01, V = 2.83, v-test (with an expected mean of 180°), N = 20) (Figure 3b, graph 1). In contrast, the differences in headings travelled by *G. unicolor* and *S. fasciculatus* in response to this treatment clustered around 0° (*G. unicolor*: μ = 353.4° ± 59.31°, *p* < 0.001, V = 3.68; *S. fasciculatus*: μ = 358.37° ± 25.58°, *p* < 0.001, V = 5.72, v-test (with an expected mean of 0°), N = 20) (Figure 3b, graph 2, 3).

As for the control for the experimental treatment (including our handling of the beetles), the changes in bearing between two exits under an unmanipulated sky was also calculated (*control*); the average change of bearings was clustered around 0° for all species (*K. nigroaeneus*: μ = 8.01° ± 42.34°, *p* < 0.001, V = 4.77; *G. unicolor*: μ = 342.76° ± 32.52°, *p* < 0.001, V = 5.14; *S. fasciculatus*: μ = 348.65° ± 26.71°, *p* < 0.001, V = 5.57, v-test (with an expected mean of 0°), N = 20) (Figure 3b, grey dotted vector). In addition, no significant difference in orientation performance was observed between the test and control conditions for *G. unicolor* and *S. fasciculatus* (*G. unicolor*: *p* = 0.61, W = 0.98; *S. fasciculatus*: *p* = 0.50, W = 1.4, Mardia–Watson–Wheeler test, N = 20). Together, these results indicate that directional information from the sun is given a greater relative weight in the orientation system of *K. nigroaeneus* compared to that of *G. unicolor* and *S. fasciculatus*.

### 3.6. The Role of Polarised Light in the Orientation System of Ball-Rolling Dung Beetles

When a polarising filter was placed above the arena aligned to the dominant e-vector direction in the open sky, then followed by a 90° rotation of the filter for the second exit (or vice versa), *S. fasciculatus* changed their heading by 82.75° ± 30.50° (N = 20) (Figure 3c, graph 3). In contrast, the change in headings recorded for *K. nigroaeneus* and *G. unicolor* clustered closer to 0° (*K. nigroaeneus*: μ = 357.64° ± 51.71°, *p* < 0.001, V = 4.22; *G. unicolor*: μ = 351.83° ± 36.02°, *p* < 0.001, V = 5.14, v-test (with an expected mean of 0°), N = 20) (Figure 3c, graph 1,2), indicating that these beetles did not respond to the 90° rotation of the polariser. This suggests that directional information from the overhead pattern of polarised skylight is given a greater relative weight in the orientation system of *S. fasciculatus* compared to that of *K. nigroaeneus* and *G. unicolor*.

When exiting twice from under a polarising filter kept in the same orientation (*control*), no significant change in direction was observed for any of the three species (*K. nigroaeneus*: μ = 355.16° ± 40.91°, *p* < 0.001, V = 4.88; *G. unicolor*: μ = 349.11° ± 38.87°, *p* < 0.001, V = 4.93; *S. fasciculatus*: μ = 333.71° ± 73.49°, *p* < 0.01, V = 4.84, v-test (with an expected mean of 0°), N = 20). In addition, no significant difference in orientation performance was observed when exiting twice under a unmanipulated polarising filter compared to exiting twice under an unmanipulated sky (*K. nigroaeneus*: *p* = 0.84, W = 0.36; *G. unicolor*: *p* = 0.93, W = 0.15; *S. fasciculatus*: *p* = 0.99, W = 0.015, Mardia–Watson–Wheeler test, N = 20), demonstrating that the addition of the polarisation filter did not have an effect on orientation performance (Figure 3c, grey dotted vector).

To further investigate the role of polarised skylight on the orientation system of *K. nigroaeneus* and *G. unicolor*, the polarising filter was again placed above the arena, but now with the sun obstructed from view. The changes in headings recorded for the two species in response to a 90° rotation of the filter now clustered around 67.36° ± 35.45° for *K. nigroaeneus*, and around 72.49° ± 36.57° for *G. unicolor* (N = 20) (Figure 3d), demonstrating that when the sun is obstructed from view, directional information from the overhead polarised light pattern is now attributed a relatively greater weight in the orientation system of these two species. 

### 3.7. The Combined Role of Sun and Polarised Skylight in the Orientation System of Garreta Unicolor and G. nitens

Given that *G. unicolor* did not turn despite a displacement of the sun or rotation of the pattern of polarised light under the open sky but did orient to an ersatz sun indoors and to a polarised light pattern in the shade, we then rotated the polariser by 90° while simultaneously mirroring the sun by 180° and shielding the real sun from the beetle’s view (Figure 3e, graph 2). To our surprise, the angular changes in bearing recorded for *G. unicolor* in response to this manipulation were not different from a random distribution (*p* = 0.70, Z = 0.37, Rayleigh uniformity test, N = 20) (Figure 3e, graph 2). It is important to note that the beetles still maintained a straight trajectory when rolling. As soon as the real sun was revealed and the polarising filter was turned back to its original position, the beetles resumed their initial direction of travel (μ = 344.67° ± 49.03°, *p* < 0.001, V = 4.23, v-test (with an expected mean of 0°), N = 20).

To further evaluate this somewhat surprising observation, we repeated this experiment on *K. nigroaeneus* and the close relative *G. nitens* (due to their experimentally frailer nature, the tiny *S. fasciculatus* would not perform under this condition, but rather flew away from the setup at any given chance). While *K. nigroaeneus* altered its heading towards a 180° turn (*μ* = 171.92° ± 85.32°, *p* = 0.02, *V* = 2.07, v-test (with an expected mean of 180°), N = 20) (Figure 3e, graph 1), the experimental outcome for *G. nitens* was similar to that of its congeneric: no change in bearing when the solar position was mirrored by 180° (*μ* = 358.37° ± 25.56°) or in response to the 90° turn of the e-vector (*μ* = 4.78° ± 49.91°), but a significant change in bearing when the two cues were rotated together (*p* < 0.001, *W* = 18.28, Mardia–Watson–Wheeler test, N = 20) (Figure 1). Similar to our findings for *G. unicolor*, the change in bearing recorded for *G. nitens* in response to simultaneous manipulation, were randomly distributed within the population (*p* = 0.44, *Z* = 0.83, Rayleigh uniformity test, N = 20). The beetles returned to their initial direction of travel as soon as the cues were rotated back to their original positions (*μ* = 8.57° ± 39.90°, *p* < 0.001, *V* = 4.91, v-test (with an expected mean of 0°), N = 20).

## 4. Discussion

In this study, we demonstrated how the orientation system of ball-rolling dung beetles, belonging to three different tribes that co-occur within the same savanna biome, attribute different relative weights to directional information during straight-line orientation.

### 4.1. Diurnal Scarabaeini Attribute the Greatest Relative Weight to the Directional Information Provided by the Sun

As with sandhoppers, monarch butterflies and birds [41,42,43], ball-rolling dung beetles can direct their straight-line movements according to directional input from a single source of light in an indoor setting (Figure 3a and [21,24,27]). Outdoors, however, the beetles are exposed to a range of celestial directional cues, including the sun, polarised skylight [44,45,46], as well as the gradients of intensity [47] and colour [48,49,50] that form across the natural sky. Therefore, if the apparent position of the real sun is changed by 180° with the aid of a mirror and a shading board, the directional information from the sun is set in conflict with that of the rest of the sky. Nevertheless, *Kheper nigroaeneus* changed its roll bearing in accordance with such an experimental displacement of the sun (Figure 3b, graph 1). A comparable response to this manipulation has also been documented for three other members of the Scarabaeini; *K. lamarcki* [21,23], *Scarabaeus ambiguus* (Boheman) [30] and *Pachysoma femoralis* Kirby [24], suggesting that the orientation system of these species attributes the greatest relative weight to the directional information provided by the sun.

Consistent with this observation, *K. nigroaeneus* did not respond to a 90° turn of an artificial, highly polarised pattern of polarised light, when presented from above in full view of the unmanipulated sun (Figure 3c, graph 1). However, as soon the sun was hidden from view, the beetles showed a clear 90° turn in response to the rotated polariser (Figure 3d, graph 1). It appears that once the sun is absent, which also naturally happens when it is obscured by a passing cloud, the distribution of the relative weight between the directional cues that remained can shift in favour of the polarised light input (Figure 3d, graph 1). The same holds true also for the close relative, *K. lamarcki* [28], where a behavioural response to the directional input from the gradients of colour and intensity can be seen when presented in isolation [28,29].

### 4.2. Sisyphus fasciculatus Attributes Greatest Relative Weight to the Directional Information Provided by the Celestial Polarisation Pattern

Neither *Garreta unicolor* nor *S. fasciculatus* changed their bearings according to the displacement of the sun (Figure 3b, graph 2, 3), indicating that the relative weight attributed to this directional cue in their orientation system is somewhat lower. This supports the results of a recent study [26], where, in contrast to *Kheper nigroaeneus, S. fasciculatus* turns in accordance with the turn of the polariser under a natural sky. Together, these findings clearly demonstrate that the smaller *S. fasciculatus* attributes the greatest relative weight to the directional information provided by the (artificial) linear pattern of polarised light. This sky-wide celestial cue is also known to play a significant role in the orientation system of other insects (locusts [51], honeybees [52] and bull ants [32]), and in some cases, even plays a dominant role (nocturnal dung beetles [21,53,54], flies [55] and desert ants [56]).

### 4.3. A Different Weighting of Directional Reference Cues in Garreta Species

To our surprise, *G. unicolor* kept to its original direction of travel both in the presence of a mirrored sun (Figure 3b, graph 2), and under a turned polariser (Figure 3c, graph 2). Only when these two cues were rotated in combination did this species demonstrate a behavioural response, which was an angular change in bearing which appeared to be randomly distributed within the population (Figure 3e, graph 2). A similar response could be confirmed in its congeneric, *G. nitens* (Figure 1), suggesting that this is a tribe-specific orientation strategy.

Due to experimental constraints, this combined manipulation of directional information from the sun and the over-head pattern of polarisation was achieved by a 180° shift in the apparent position of the sun in combination with a 90° rotation of the polarisation pattern. The outcome of these manipulations was that the position of the two cues were not only changed in relation to the unmanipulated gradients of intensity and colour that spans the sky, but also in relation to each other. This drastic and multi-angular change in directional input could potentially cause the beetles to simply re-set their roll bearings, effectively contributing to the random changes in bearings displayed by *G. unicolor* (Figure 3e). However, this conjecture can be refuted, as the beetles faithfully returned to their initial bearings as soon as the cues were returned to their initial positions (Figure 3e, grey dotted vector, graph 2). Additionally, when tested under the same multi-conflict paradigm, *K. nigroaeneus* showed a clear and directed response. Attributing the greatest relative weight to the sun, these beetles simply continued to follow the angular displacement of this cue also under this experimental condition (compare Figure 3b,e). One possibility is of course that in our experiments with *G. unicolor*, each beetle followed an individual strategy; some turned 180° according to the sun, some 90° according to the overhead polarisation and some followed the stable gradients of intensity and colour. This is, however, unlikely, as we would then have expected to see a different and much more varied response when these cues were manipulated on their own (see Figure 3b,c). The random spread of changes in bearings observed for the *Garreta* sp. rather points to a more even weighting of directional information where the combined directional information in this artificial cue conflict experiment, results in a weak directional signal. While the beetles were still able to exit from the centre of the arena along straight paths, small, individual differences in the weighting of cues could now be seen in large differences in angular change. It would have been interesting to evaluate this theory further by testing the same beetle repeatedly before rotating all cues back to their initial positions again, but this was unfortunately not within the scope of this study.

While our results do not reveal the precise nature of the orientation strategy of the *Garreta* species, we can still conclude that the heading direction networks of our three test species process the directional information provided by the sky somewhat differently; *K. nigroaeneus* preferentially steers according to the sun, *S. fasciculatus* with the pattern of polarised light, and *G. unicolor* (and *G. nitens*) does not attribute a greater relative weight to either of these cues.

### 4.4. Compass Cue Integration and Its Relation to the Visual Ecology of Ball-Rolling Dung Beetles

Given that a navigator can reliably perceive and analyse directional information provided by the sun and its pattern of polarised light, neither of these cues should be inherently more reliable for orientation than the other. We previously showed that ball-rolling beetles that attribute the greatest weight to directional information provided by the sun, are equally well directed in its absence when an alternative cue is available [23,28]. This holds true also for ants [56], monarch butterflies [9] and fruit flies [55]. It is further important to note, that the three tribes of dung beetles tested in this study—each attributing a different relative weight to the sun and the celestial polarised light pattern—all orient with the same precision under a clear, open sky (Figure 2b, grey mean vector (*control*)). Taken together, this indicates that the directional information provided by the sun or polarised light in the photon-rich African sky can (i) support orientation with the same precision, and (ii) be processed with comparable accuracy by the visual system and heading direction network of the Scarabaeini beetles. This is most likely also the case for the Gymnopleurini (species *G. unicolor* and *G. nitens*), that do not seem to employ differential weighting to any of the celestial cues tested.

Even though our test species are active within the same bioregion, *K. nigroaeneus* and *G. unicolor* were found actively foraging in the open habitat (Figure 1a), while the smaller *S. fasciculatus* rather foraged for dung within the closed habitat (Figure 1b). In this habitat, with tall grass and a high density of trees, the sun will be frequently obstructed from view, while a wide-field cue, such as the celestial polarised light pattern, will remain visible through any overhead vegetation [45,57,58]. This is also the cue attributed the highest directional weight in the orientation system of *S. fasciculatus* [26]. While it would have been preferable to explore the orientation strategy of additional species within the tribe Sisyphini from a different visual habitat, this unfortunately proved impossible as the species available to us (*Sisyphus manni* [59] and *Sisyphus seminulum* [60]) are so small (pronotum width: 3–5 mm) and timid, that not even the most experienced beetle experimentalist could coerce them into performing in our experiments. Still, our limited results from this tribe again suggest that they afford the greatest weight to the most consistent source of celestial directional information in their cluttered habitat [7], a strategy also found in ants [18,19].

As day turns into night, the visual world changes drastically, most notably in the decrease in light intensity [61]. At this time of the day, visually driven orientation systems need to capture as much light as possible. One common way to meet this challenge is by an increase in eye size [62,63], but it is also interesting to note that some neurons within the heading direction network of the desert locust have a higher absolute sensitivity to polarised than to unpolarised light [64]. The larger eyes of the nocturnal, open habitat Scarabaeini beetles (*Scarabaeus satyrus* and *S. zambesianus*), possess a large dorsal rim area (DRA) (the polarisation sensitive region known to detect polarised light in insects [65,66,67]) and rely on directional information from the polarised skylight above that provided by the moon itself [25,53,54]. In contrast, the diurnal *K. lamarcki*, which is active in the same habitat, only possess a single dorsal row of polarisation-sensitive ommatidia (Dacke *unpublished data*). The orientation systems of beetles active under more challenging light conditions—in the dark or under vegetation canopies—thus seem well adapted to their respective visual ecologies.

It is interesting to note that differences in external eye morphology between the ball-rolling beetles are more pronounced between the three tribes, than within the tribes themselves [68] (Figure 2). As representatives of their respective tribes ([69,70,71]), *S. fasciculatus* and *G. unicolor* possess a more oval-shaped dorsal eye compared to that of *K. nigroaeneus*, where the dorsal eye of the medium-sized *G. unicolor* is proportionally smaller than that of *K. nigroaeneus*, while the small and spindly *S. fasciculatus* has the smallest dorsal eye of the three (Figure 2b). If we would assume that the small and narrow dorsal eyes of the smaller species also have a smaller visual field [72,73,74,75,76], the heading direction networks of the narrow-eyed Sisyphini and Gymnopleurini could possibly benefit from a sky-wide orientation signal, such as the celestial polarisation pattern, rather than using the position of a single light source. These inter-tribal differences might be an additional influence on how species within each tribe weight the sources of directional information they can reliably use.

In conclusion, despite exhibiting different body size, eye size and morphology, dung beetles nevertheless manage to solve the challenge of straight-line orientation by weighting visual cues that are particular to the habitat in which they are found. This system is however dynamic, allowing them to operate equally well even in the absence of the cue given the greatest relative weight by the particular species.

## Figures and Tables

**Figure 1 insects-12-00526-f001:**
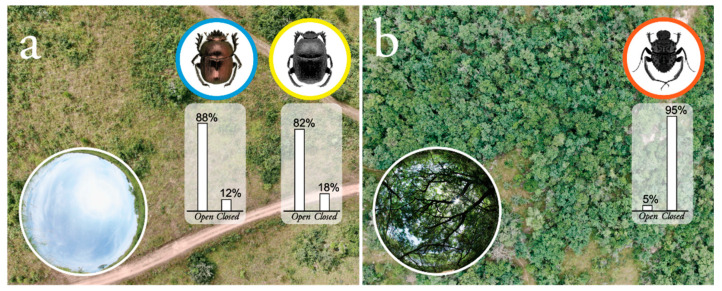
Dung beetles from three tribes of ball-rollers and the bioregions they inhabit. Beetles from three tribes of ball-rollers (*blue-bordered image*: Scarabaeini; *yellow-bordered image*: Gymnopleurini; *red-bordered image*: Sisyphini) were collected within the same savanna biome. *K. nigroaeneus* and *G. unicolor* were predominantly found actively foraging in the open habitat (**a**) and *S. fasciculatus* predominantly foraged within the closed habitat (**b**) of the same bioregion. A histogram, illustrating the percentage of individuals found in the open and closed habitat over three consecutive sampling days, is presented below each respective beetle image. A 180° view of the sky from the ground perspective of the beetle is included at the bottom of each panel.

**Figure 2 insects-12-00526-f002:**
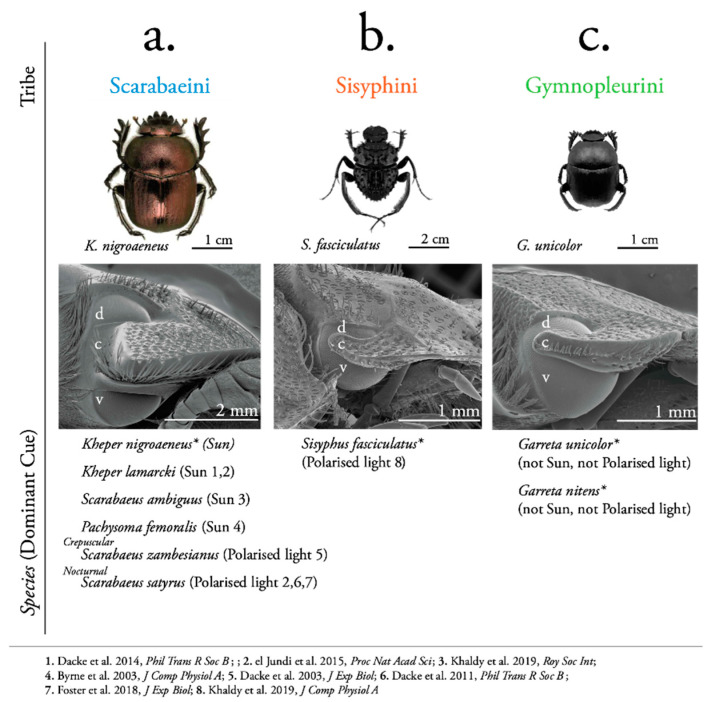
Habitus, eye design and dominant orientation cue in three tribes of ball-rolling dung beetles. Diurnal species within the tribe Scarabaeini (**a**) attribute greatest relative weight to directional information provided by the sun during straight-line orientation. In contrast, two nocturnal species from the same tribe, as well as the smaller, diurnal *Sisyphus fasciculatus*, from the tribe Sisyphini (**b**), rely predominantly on polarised skylight for directional information. The underlying weighting strategy for straight-line orientation within the tribe Gymnopleurini (**c**) differs from that previously mentioned, where neither directional information from the sun nor the polarisation pattern dominates the output from its compass network. As can also be noted from our test species, Sisyphini are generally much smaller than Gymnopleurini, which in turn are smaller than Scarabaeini. The relative eye sizes across the three tribes follow the same pattern, but they differ in shape. The canthus (*c*) completely separates the roughly equal sized dorsal (*d*) and ventral (*v*) eyes of the Scarabaeini, while the dorsal portion of the eye of the Gymnopleurini and Sisyphini is only partially separated and much smaller than the ventral part. Species tested in this study are indicated by an asterisk (*).

**Figure 3 insects-12-00526-f003:**
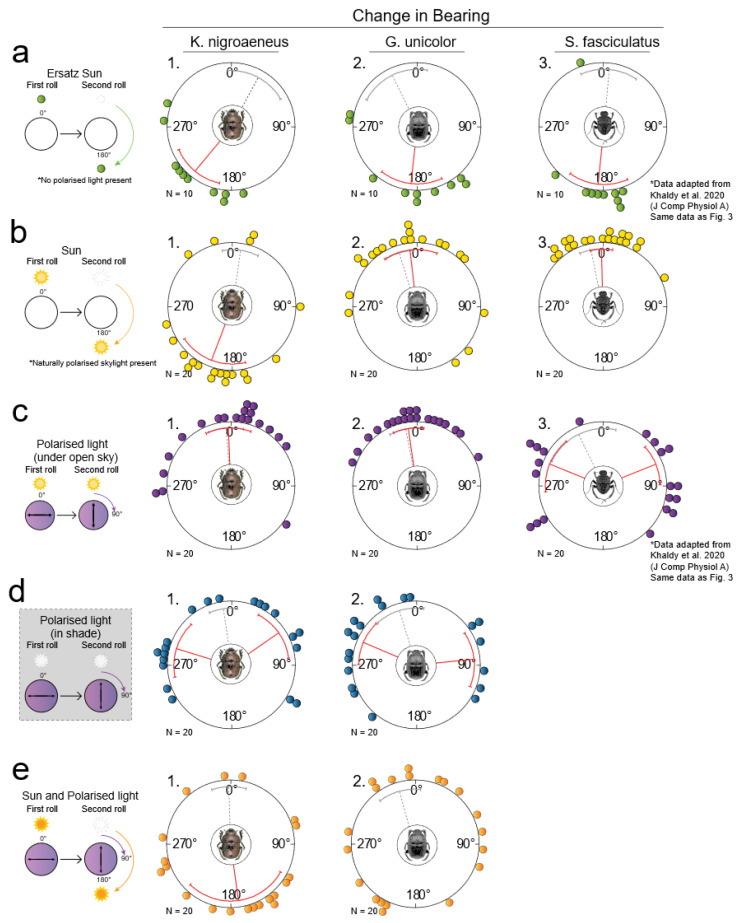
Response to directional change of compass cues. Three diurnal dung beetle species (from left to right; *Kheper nigroaeneus*, *Garreta unicolor* and *Sisyphus fasciculatus*) were allowed to roll balls out of a circular arena in a darkened room (**a**) or outdoors under the open sky (**b**), or with a polarisation filter placed above the arena with the sun visible (**c**,**e**) or with a polarisation filter placed above the arena with the sun shielded from view (**d**). Once the beetle had reached the periphery of the arena, it was removed from its dung ball and placed back in the centre alongside its ball. At this time, the apparent position of the ersatz sun ((**a**), *green arrow*) or the real sun ((**b**), *orange arrow*) was switched by 180°, the apparent e-vector direction was turned by 90° using a polarisation filter ((**c**,**d**), *purple arrow*) or the position of the sun was changed by 180° while simultaneously turning the apparent e-vector direction by 90° ((**e**), *orange arrow:* sun; *purple arrow*: polarisation filter). The beetle was then allowed to exit the arena a second time. The absolute angular difference between the first and the second exit angle represent the response to the treatment (*test*). (**a**): With the ersatz sun switched by 180°, all three species changed their bearings in accordance with this angular change (red vector, all graphs); (**b**): with the sun mirrored by 180° outdoors, only *K. nigroaeneus* showed a significant change in bearings in response to this manipulation; (**c**): with the e-vector turned by 90° under a clear sky, only *S. fasciculatus* responded significantly by a change in exit bearings approaching 90°; (**d**): with the sun shielded from view, a significant change in bearings could also be elicited in *K. nigroaeneus* and *G. unicolor*; (**e**): with the sun mirrored by 180° and the polarisation pattern e-vector turned by 90°, *K. nigroaeneus* changed its exit bearings in a similar fashion to when only the sun was mirrored by 180° (see graph 1 in (**b**)), while the changes in exit bearings for *G. unicolor* were randomly distributed within the population (graph 2). Beetles were then allowed to roll a third time, with the manipulated cue(s) moved back to its/their initial position. The mean angular difference between the first and second exit (*test*), and the first and third exit (*control*), is represented by a red solid vector and a grey dotted vector, respectively, in each graph. Error bars represent one circular standard deviation. The data presented for *S. fasciculatus* in (**a**) and (**c**) (graph 3, respectively) were adapted from Khaldy et al., 2020 [26]. For detailed data see Appendix A.

## Data Availability

All data supporting reported results can be found in Appendix A.

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
