# Peer review of "Compass Cue Integration and Its Relation to the Visual Ecology of Three Tribes of Ball-Rolling Dung Beetles"

_insects, 2021, doi:10.3390/insects12060526_

Round 1

Reviewer 1 Report

In this comparative article Khaldy et al., explored the relative contribution of celestial visual inputs in the dung beetles' choice of orientation.

MINOR COMMENTS:

At least in some cases 'relative WEIGHT of cues' perhaps sounds better than 'relative WEIGHTING of cues', which the authors wrote.

The concept of 'tribe' should be made more clear either in the abstract or in the intro

Unnecessary use of '-' should be checked, e.g., in 'sun-lit', 'moon-lit', 'pin-point', etc. a hyphen is not typically used.

The authors' earlier article Khaldy 2020 (in bibliography) is often mentioned in the text as Khaldy 2019.

In fig 2 a and b, it might be useful to explicitly state the polarization status, (even if the light is not polarized).

To represent the orientation vector, the magnitude (r) and the angle (theta) are generally shown in circular statistics. The scalar magnitude of the vectors all seem to be 1 in the graphs of Fig 2. Shouldn't the value of 'r' be clearly represented?

Could the authors point to any previous data to indicate that indeed, depending on the microhabitat, the reliability/consistency/robustness of polarisation versus sun-position cues systematically vary.

The discussion part may be shortened. Although what putative orientation strategy does Garreta use - even if it's a conjecture at this point - may still be useful to discuss briefly.

Reviewer 2 Report

The article “Compass cue integration and its relation to the visual ecology of three tribes of ball-rolling dung beetles” by Khaldy and colleagues provides very interesting insights into different sky-compass navigation strategies of different species of dung beetles, set in the context of their natural habitats and the visual environment they perceive in these. The species chosen are very appropriate for the comparison, and the core experiments performed highly instructive. I have two major concerns with the current presentation of the study (apart from a number of smaller suggestions) that I am confident the authors can address in a revision of their manuscript.

1) Data analysis and statistics: I am not convinced that appropriate statistical tests were used for both the abundance per habitat data and the circular orientation data. In both cases I would also suggest that potential confusions of the reader might be alleviated by explaining the data analysis and statistics procedures in some more detail in the results section (even if they have been published in a similar form before).

Habitat abundance scores: to start with, I am not sure about the structure of this data, as it is presented as fractions, which add up to 100%. Does this represent the fraction of beetles of this species which were captured in either traps in open and closed habitats? In which case I don’t understand how it is possible to compare these with a paired t-test, rather than a statistical test appropriate for fraction comparisons (i.e. binomial test). If there were multiple samples of each habitat that would allow for some pairwise comparison of multiple samples, it would be helpful to indicate the sampling errors in the plots, and even then a t-test would likely not be an appropriate measure for statistical comparison, as fractional data rarely fulfils its assumptions (though if it does, it should be clarified how in the methods section).

Circular orientation data: while I could not find further information on the topic in the methods section, it looks like the exit headings or orientation errors of the dung beetles were converted from a 0-360° scale to a 0-180° scale – as I gathered from the descriptions in the methods that the experiments were conducted in circular arenas, while the plots only show values from 0 to 180°. The methods state “Angular change was calculated as the difference in bearing between the first and second exit (test), or first and third exit (control).“ (l.187-188), suggesting that in the first analysis step, the original configuration was preserved (as differences could be both positive and negative).

I am puzzled by this conversion, because it does not represent the population orientation of the dung beetles appropriately, as soon as there is some spread in the beetles orientation around the expected value. Let’s take the scenario where beetles were tested multiple times under the same condition, and the population clusters around an orientation error or change of 0° but with some spread. There would be some values between 270-360 (or -90 and 0) and some at 0-90, but on a population level the mean orientation vector would be close to 0°. Now, converting these orientations to 0-180° would result in all orientation errors being larger than 0°, and the population vector therefore also being centred not close to 0°, but at a distinctly larger value. And this effect can be seen in Fig. 2. In essence, it is impossible to say after this conversion whether the beetles had a bias for orientations around 45° (in conditions where they did not follow the 180° turn of the stimuli), or whether the population was centred on 0°, but the conversion to 0-180° generated this biased population vector. And similar for the conditions where the animals did change orientation with the external stimuli, resulting in population vectors around 120°. What is even more concerning is that the V-test applied to statistically assess the data assumes expected values of 0° and 180°, but these are not appropriate for the converted angular values, only for the 0-360° data range, rendering the statistical tests inappropriate. Moreover, after conversion to (0°,180°) these data are no longer “circular”, i.e. data that fall on the perimeter of a unit circle (Batschelet, 1981), since they can only fall on a semicircle. They therefore cannot form a von Mises distribution (H1 for a Rayleigh or v-test) or a circular-uniform distribution (H0 for these tests).I would therefore urge the authors to use their original data range of 360° for both the visualisation of the data and statistical analysis – or to use other statistical tests for data analysis appropriate for the specific data type used

2) Framing of the study in terms of visual ecology: I really appreciate the authors’ focus on investigating the navigation strategies of dung beetles in the context of their visual ecology. However, given that this is in the title of the paper and the scope outlined in the introduction, the results could have been analysed and interpreted with a much stronger focus on this question. In the following I outlined a number questions and suggestions connected to this point:

- Methods 2.3: By which criteria was an open and a closed habitat defined? This should be described in more detail. For example, in the discussion, the closed habitat of S. fasciculatus is characterised through tree canopies and high grass – does that mean the grass was short in the open habitat? Where there any trees, and if so, what was the criteria that set the two habitats apart?

- In the discussion section (in part. 457-471) the authors discuss very convincingly why S. fascifulatus might benefit from weighing polarisation cues stronger, given the canopied habitat they live in. However, I am missing a more detailed discussion on the orientation strategies and visual ecology of the other two species: both K. nigroaeneus and G. unicolor live in an open habitat, but their orientation strategies differ. How might this be related to their specific visual ecology, if at all? Why might it be beneficial to rely on the sun in an open habitat? Given that clouds could shadow the sun, would it not also be a beneficial strategy for open habitat dwellers to rely on the polarisation pattern, as that seems to be “fool-proof”? Or is there an inherent benefit of using the sun if possible? To really tie the results in with the scope of the title and introduction, it would be helpful to discuss these questions in the discussion section.

- the comparison of eye morphology provided a very valuable addition to the behavioural data – however, I did not understand why it was only discussed in the discussion section, and almost as an afterthought, rather than integrated into the results section, and treated more quantitatively. The comparison of dorsal eyes as “narrower” (l.489) left me wondering, what the authors meant by that (narrower in which dimension, and in absolute terms, or relative eye size), and wishing that the eyes of the three species were compared in a more quantitative way (absolute eye size, absolute diameter, and size of the dorsal rim area). In the current form, little can be gained from the three eye images of the three species. Especially given the intriguing results of different orientation strategies with respect to the sun and polarised light, comparing the relative size of the DRA and rest of the dorsal eye for the three species would be highly instructive in appreciating the adaptations of the beetles for their different visual ecologies and visual behaviours – as the authors nicely highlight for the comparison between diurnal and nocturnal dung beetles (l. 480-483).

Further suggestions and comments:

  • 326-329 if I understand correctly, this represents an unnatural scenario to the beetle, as the sun and polarised light would not naturally turn in this way (but both by the same angular distance). It might be helpful just to mention this briefly so all readers immediately understand.
  • 421-426 should this effect not also be visible in the beetles’ walking tracks, or exist times – as the authors have so nicely demonstrated for disoriented beetles in previous studies? It would be very instructive to add such an analysis, even for a subset of beetles if possible, as it would strengthen the argument that the beetles do not just follow individual strategies of orientation in this scenario, but are actually disoriented.
  • 426-428 I am afraid to say that I could not follow the authors explanation for how the heading direction network and sensory noise could lead to the observed random orientation directions – since this is a key point in helping to make sense of the surprising behavioural results, I would encourage the authors to spell out this point in some more detail. For example: what dot hey expect the output of the orientation network to be for this cue condition, and what role does sensory noise take in shaping the behavioural responses?
  • 472 – is this the wrong way around, and should be “as day turns into night”? otherwise I do not understand the reference to the decrease in light intensity?
  • 473 which time of day is meant here?
  • 474-475 I am not sure this statement is correct in its generality: there are certainly other visually driven orientation systems that use mechanisms to increase sensitivity other than the polarisation system (or rather, have no connection to the polarisation system at all).
  • 475-476 I cannot quite follow the argumentation here: how does the specificity for polarised light of the LoTu1 neurons in locusts relate to adaptations for increased sensitivity or photon capture (in dim light)?
  • L488 – what is meant by “fair” representatives?
  • 493 – “Assuming that a smaller compound eye has a smaller visual field [70–74],” I am not convinced that this can be assumed generally – as it depends on the curvature of the eye, in combination with the overall area the eye covers – a view that I would argue is also supported by the cited references. A perfectly spherical compound eye of the same angular extension has the same field of view, whether it has a large or small diameter.
  • 494-495 – or their eyes might have a narrower field of view (if they do) because their orientation strategy does not require a whole-sky field of view? As in, it might be hard to tell which one is the hen and which one the egg.

Reviewer 3 Report

Very few suggestions within the attached PDF

Round 2

Reviewer 2 Report

I would like to thank the authors for their detailed responses to all my queries, and the easy to follow explanation of what was changed in the manuscript.

They addressed all my questions and suggestions to my full satisfaction.

The only thing I noticed to improve before publication is to explain the measures in which the eye size data is given (mean ± s.d.)? Apologies if I have missed that if it was explained!